# Harnessing C/N balance of *Chromochloris zofingiensis* to overcome the potential conflict in microalgal production

Han Sun[1,2,3], Yuanyuan Ren[1,2,3], Xuemei Mao[1,2], Xiaojie Li [1,2], Huaiyuan Zhang[4], Yongmin Lao[1,2] & Feng Chen [1,2✉]

Accumulation of high-value products in microalgae is not conducive with rapid cell growth, which is the potential conflict in microalgal production. Overcoming such conflict faces numerous challenges in comprehensively understanding cell behavior and metabolism. Here, we show a fully integrated interaction between cell behavior, carbon partitioning, carbon availability and path rate of central carbon metabolism, and have practically overcome the production conflict of *Chromochloris zofingiensis*. We demonstrate that elevated carbon availability and active path rate of precursors are determinants for product biosynthesis, and the former exhibits a superior potential. As protein content reaches a threshold value to confer survival advantages, carbon availability becomes the major limiting factor for product biosynthesis and cell reproduction. Based on integrated interaction, regulating the C/N balance by feeding carbon source under excess light increases content of high-value products without inhibiting cell growth. Our findings provide a new orientation to achieve great productivity improvements in microalgal production.

[1] Institute for Advanced Study, Shenzhen University, Shenzhen 518060, China. [2] Shenzhen Key Laboratory of Marine Microbiome Engineering, Shenzhen University, Shenzhen 518060, China. [3] Institute for Food & Bioresource Engineering, College of Engineering, Peking University, Beijing 100871, China. [4] School of Agricultural Engineering and Food Science, Shandong University of Technology, Zibo 255000, China. ✉email: sfchen@szu.edu.cn

Bio-based economy is urgently being driven by the changing global climate and the increasing demand of renewable biofuels and oleochemicals[1-3]. Microalgae, photosynthetic cell factories with high photosynthetic efficiency, are essential for capturing and recycling the global carbon dioxide, and potentially convert it into biomass over time and distance. In addition, microalgae can be exploited in wastewater treatment and biofuel production due to their advantages of nitrogen and phosphorus loading, no competition with food production, high oil content, and rapid growth rate[4,5]. Alternatively, microalgae are also considered as the promising cell factories producing profitable natural pigments and lipids in food fields, and offsetting the cost in various treatments during cultivation and harvesting[6]. Consequently, environmental issues and food revenues make microalgal production a matter of overall consideration. One of the most obvious examples is the conversion of process-compatible products always negatively affects cell growth. Recently, optimized culture strategies and genetic engineering approaches have been developed for alleviating the potential conflict between biomass concentration and its valuable products[2,7]. However, the obstacle still remains as productivity of biomass and associated lipids and pigments is controlled by a set of complex but poorly understood environmental and genetic factors[1]. Furthermore, considering the challenges of applicability and validity under Algae Raceway Integrated Design, microalgal production is needed to coordinate environmental conditions in agricultural and industrial processes[8,9], and thus commercialization are very different. To efficiently address environmental issues with considerable economic benefits, understanding of underlying relationship between the conditions and cell behavior is essential.

Microalgae excel in adapting to environment or saving energy for survival under various environmental conditions[10-12]. Among the condition factors, light and nutrient have been reported to take charge in the important physiological and morphological responses[13]. Though light is essential for photosynthesis, the solar-to-biomass conversion efficiency is <6% for most photosynthetic organisms[4,14]. Up till now, light intensity, light quality, and light cycle have been tested in order to confirm their best implementation in cell growth, reproduction and accumulation of high-value products[15]. On the other hand, nutrient is involved in carbon and nitrogen metabolism, where modes of repletion, starvation, recovery and recycle are considered as valid strategies for increasing the productivity of biomass and high-value products[16]. Furthermore, coordinating advantages of light and nutrient brings out various powerful strategies for microalgal cultivation such as light-increment culture, flashing-light culture, mixing-light culture, fed-batch culture, gradient-fed culture, semi-continuous culture, and nutrient-starvation stress culture[17-20]. However, since most reported cultivations are carried out in a black-box model through a single chemical reaction of nutrient into biomass, precision and adaptability of the strategies proposed above are limited in the realistic cultivation. In this case, global understanding between cell behavior and cell metabolism responding to a certain environmental factor is needed for achieving a higher biomass value.

In a cell factory, dynamics of biomass component and metabolite analysis have revealed that light and nutrient cause compositional shifts through carbon partitioning and central carbon metabolism[18]. Variations in light and nutrient would lead algal cells to grow at a steady-growth state or redistribute carbon sinks for survival. For instance, previous study stated sufficient nutrient preferably increased the biomass, while excess light and nitrogen starvation favored in the accumulation of lipid and carotenoid by elevating pyruvate availability[17,18]. Consider excess light has a great potential to promote accumulation of carotenoid and polyunsaturated fatty acids (PUFAs). It concludes that elevated intracellular carbon availability is beneficial for biosynthesis of

lipid and secondary metabolite[18,21]. However, given the fact that elevated carbon availability under stress condition is at the expense of cell growth[18], it is essential to systematically investigate the relationship between carbon efficiency and path rate of the products, since efficient exogenous carbon ensures the cell reproduction and sufficient intracellular carbon availability helps to gain revenues from lipid and secondary metabolites. At the molecular level, tricarboxylic acid (TCA) cycle and synthetic pathways of lipid and carotenoid share pyruvate and acetyl-CoA as carbon precursors, which can be consumed at a fast rate once the synthetic genes are upregulated[4]. However, the path rates under different environmental factors are seldom studied, limiting the metabolic regulation to suppress or remove the pathways with high-energy consumption. Furthermore, no report has focused on the significant influence of exogenous carbon on cell behaviors and metabolism. Exogenous carbon molecule is the key factor to link cell growth and product accumulation through its complex influence on carbon availability and path rate of carbon metabolism[4]. When tolerating stress conditions, algal cells commonly pave ways for lipid accumulation through protein degradation, which provides carbon skeletons, energy molecules (adenosine triphosphate, ATP), and reducing power (reduced nicotinamide adenine dinucleotide phosphate, NADPH)[4]. In addition, as cellular protein stoichiometry is largely dependent on nitrogen availability[22], the supplied carbon is then linked to the carbon/nitrogen (C/N) stoichiometric ratio, which is reported to range from 6:1 to 15:1 as a result of acclimatization and subsistence[23,24]. Therefore, a comprehensive understanding of the relations among nutrient, light, and C/N balance at the molecular level will no doubt deliver great productivity improvements, and then blaze a new frontier in microalgal production for the highest revenue.

The objective of the current study is to comprehensively understand the influence of light and nutrient on microalgal production in carbon partitioning at the molecular level. First, a kinetic model is established to explore the changes of carbon partitioning under different light intensities, followed by a determination of photosynthetic characteristics. Then, transcriptomics and $^{13}$C tracer-based metabolic flux analysis ($^{13}$C-MFA) are applied to explain the carbon behaviors in central carbon metabolism. In combination with light, exponential fed-batch cultures providing different C/N ratios are designed to explore complex interactions of carbon source, carbon availability and path rates of central carbon metabolism. Such interactions are finally described through metabolic flux analysis and targeted metabolites of precursors for lipid and carotenoid biosynthesis. The results of this study will potentially overcome the conflicts between biomass concentration and high-value product yield, and make it possible to accurately design cultivation processes of microalgae for the highest revenue.

## Results

**Carbon distribution impacted biomass accumulation.** As light is an essential source for photosynthetic activity, its change in intensity potentially influences the biomass concentration[15]. As shown in Fig. 1, the effect of light intensity on biomass was evaluated in the aspect of carbon partitioning. Comparing with heterotrophic culture, mixotrophic culture enhanced biomass concentration as the intensity increased to $150\,\mu E\,m^{-2}\,s^{-1}$, and subsequently reduced at $300\,\mu E\,m^{-2}\,s^{-1}$. In addition, intensity at $50\,\mu E\,m^{-2}\,s^{-1}$, as the most suitable light intensity, induced the highest growth rate with the maximal biomass at 72 h. The previous study suggested excess light could restrain cell growth by changing carbon partitioning and limiting photosynthetic efficiency[18]. As shown in Fig. 1b, protein content tended to decline as the time

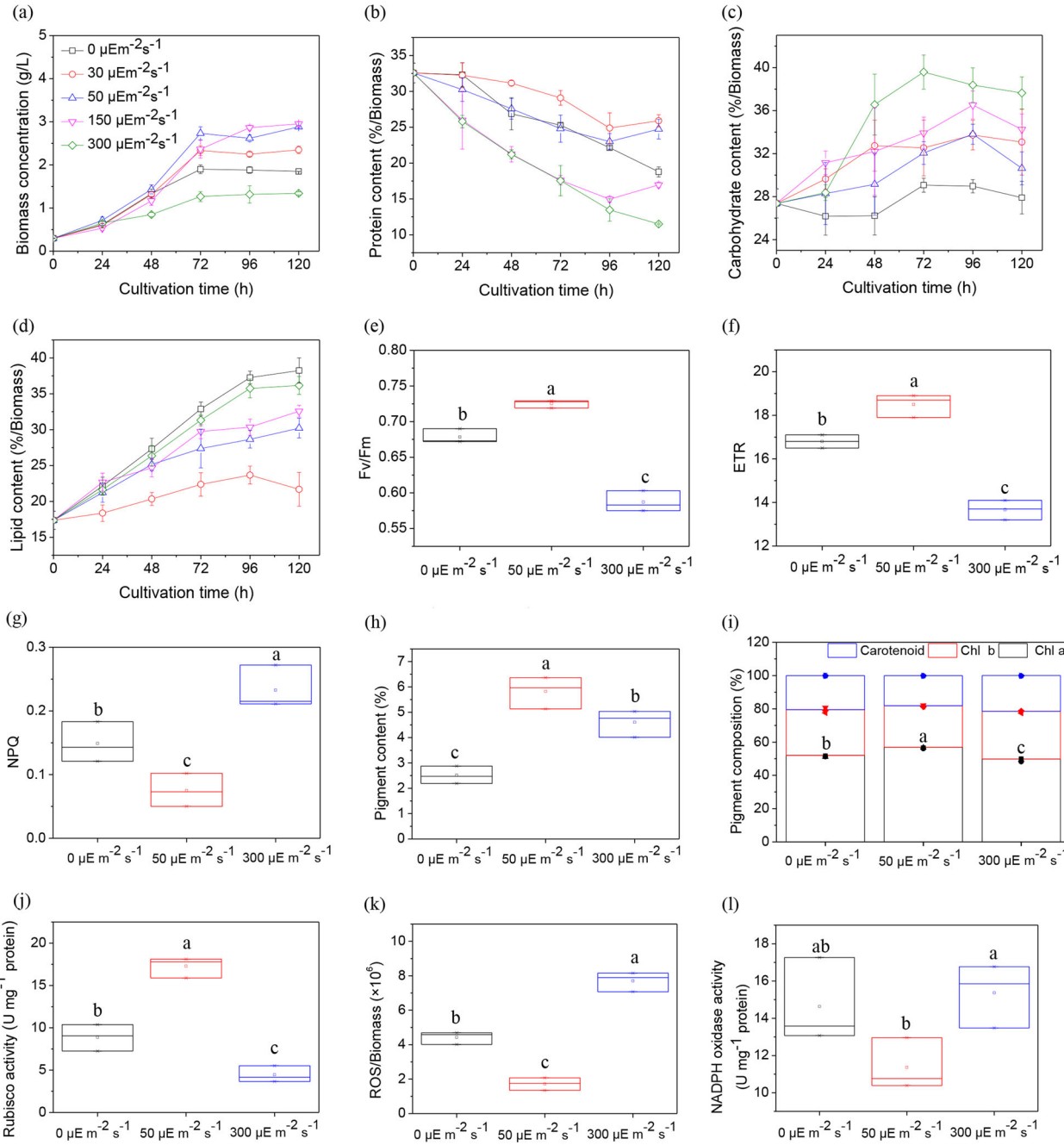

**Fig. 1 Effect of light intensity on cell behavior.** Growth profiles (**a**) and changes of carbon partitioning in protein (**b**), carbohydrate (**c**), and lipid (**d**) under different light intensities. Photosynthetic characteristics of $F_v/F_m$ (**e**), ETR (**f**) and NPQ (**g**), pigment content (**h**) and composition (**i**), Rubisco activity (**j**), ROS level (**k**), and NADPH oxidase activity (**l**) under different light intensities. The metabolites: Chl *a* chlorophyll *a*, Chl *b* chlorophyll *b*. Post-hoc comparison, different superscript letters indicate significant difference ($p < 0.05$), $N = 3$.

extended, which was more obvious under excess light. Accordingly, carbohydrate and lipid contents were increasing as the culture time went on, and reached the highest at $300 \, \mu E \, m^{-2} \, s^{-1}$. The decreased protein would lead to growth retardation or even cell death[4]. Interestingly, although protein degradation at heterotrophic culture was slower than those at 150 and $300 \, \mu E \, m^{-2} \, s^{-1}$, its lipid accumulation rate was the highest among the cultures. The changes of carbon partitioning under dark and light conditions followed different rules.

Therefore, a kinetic model was designed to explore the dynamic changes of carbon partitioning. As shown in Table 1, values of $D_P$, $\alpha_C$ and $Y_{CX,max}$ were all the highest at $300 \, \mu E \, m^{-2} \, s^{-1}$, which

suggested excess light induced carbon flux from protein to carbohydrate and then to lipid. However, $D_P$ in dark was close to that at $50 \, \mu E \, m^{-2} \, s^{-1}$, whereas $\alpha_C$ and $Y_{CX,max}$ were lower than those at $50 \, \mu E \, m^{-2} \, s^{-1}$, which revealed carbon flux was oriented from protein into lipid directly as the time extended under heterotrophic culture. Therefore, elevated carbon availability for lipid biosynthesis was mainly due to the degraded protein molecules both under excess light and darkness. Moreover, $Y_{PX,max}$ and $\mu_{glucose}$ were at a higher level under heterotrophic culture, which suggested that direct connection of protein and lipid might be beneficial for the conversion of glucose into protein. Therefore, numerous researches suggested that heterotrophic

                3

**Table 1 Coefficients of the model for carbon partitioning.**

| Coefficient | $0\,\mu Em^{-2}s^{-1}$ | $50\,\mu Em^{-2}s^{-1}$ | $300\,\mu Em^{-2}s^{-1}$ |
|---|---|---|---|
| $Y_{PX,max}$ ($g\,g^{-1}$) | 0.330 | 0.326 | 0.317 |
| $D_P$ ($\times 10^{-2}\,h^{-1}$) | 0.139 | 0.137 | 0.310 |
| $Y_{CX,max}$ ($g\,g^{-1}$) | 0.286 | 0.353 | 0.396 |
| $\alpha_C$ ($\times 10^{-2}\,h^{-1}$) | 0.604 | 1.130 | 4.424 |
| $\mu_{L,max}$ ($h^{-1}$) | 0.047 | 0.046 | 0.048 |
| $\mu_{glucose}$ ($h^{-1}$) | 0.048 | 0.044 | 0.017 |

cultivation was an efficient and sustainable approach for some microalgae to produce lipids[4,6].

Photosynthesis begins with the activation of assimilative pigment centralized by light harvesting antenna complex. Its activity ensures a higher $CO_2$ uptake rate and biomass concentration[4]. As shown in Fig. 1e–g, $F_v/F_m$ and ETR under darkness and excess light were both lower than those under suitable light intensity. As $F_v/F_m$ is a stressing indicator and ETR represents electron transfer, their decreases suggested darkness and excess light inhibited photosynthesis[18]. NPQ under darkness ($p = 0.041$) and excess light ($p = 0.005$) was significantly higher that revealed more energy was quenched as thermal dissipation. Accordingly, reported researches revealed that excess light damaged the photosystem of *Chlamydomonas reinhardtii* and heterotrophic culture limited the photosynthesis of *C. zofingiensis*[18,25]. In addition, as 0.03% $CO_2$ (V/V) in air was not sufficient for the rapid growth under mixotrophic culture, the photosynthetic efficiency could not achieve the maximum due to the tough limitation by $CO_2$[26]. Besides, the exogenous glucose was also revealed to trigger chloroplasts degradation and decrease photosynthesis of *C. zofingiensis*[25,27]. By comparing photosynthetic characteristics under darkness and excess light, light intensity at $300\,\mu E\,m^{-2}\,s^{-1}$ created a more severe stress condition that largely limited photosynthetic efficiency, as its $F_v/F_m$ was lower ($p = 0.001$) and NPQ was higher ($p = 0.035$) than those at $0\,\mu E\,m^{-2}\,s^{-1}$. In addition, Rubisco-bisphosphate carboxylase/oxygenase (Rubisco) is the first enzyme involved in Calvin cycle, which largely determines the photosynthetic carbon assimilation. As shown in Fig. 1j, Rubisco activity at $0\,\mu E\,m^{-2}\,s^{-1}$ ($p = 0.002$) and $300\,\mu E\,m^{-2}\,s^{-1}$ ($p < 0.001$) was reduced significantly compared with that at $50\,\mu E\,m^{-2}\,s^{-1}$, suggesting the photosynthetic carbon assimilation was limited. Furthermore, the limited $CO_2$ potentially reduced the assimilative capacity under the mixotrophic culture[26]. Therefore, the speediest cell growth under $50\,\mu E\,m^{-2}\,s^{-1}$ could be attributed to the higher $\mu_{glucose}$ and the highest level of Rubisco activity. On the other hand, $\mu_{glucose}$ and Rubisco activity were the lowest at $300\,\mu E\,m^{-2}\,s^{-1}$, which revealed that elevated carbon availability for lipid biosynthesis was mainly due to the conversion of carbon partitioning under mixotrophic culture with excess light.

Light harvesting antenna complex in microalgae is commonly comprised of about 80% chlorophyll and 20% the rest pigments, which account for around 5% of biomass[4]. Pigment content and composition influence photosynthetic efficiency and cell growth. As shown in Fig. 1h, i, excess light and darkness suppressed pigment accumulation. As chlorophyll *a* and chlorophyll *b* are major light harvesting pigments, their decreases in pigment composition could also cause a limited photosynthetic efficiency[28]. In addition, energy for cell metabolism under heterotrophic culture is from respiratory action[29]. However, $O_2$ could be converted into reactive oxygen species (ROS) by electron transfer of NADPH oxidase[30]. The intracellular ROS at $0\,\mu E\,m^{-2}\,s^{-1}$ ($p = 0.001$) was significantly higher than that at $50\,\mu E\,m^{-2}\,s^{-1}$ with active NADPH oxidase. Therefore, heterotrophic culture had

a great potential to produce large amounts of NADPH and achieve high ROS levels, which were beneficial for lipid biosynthesis. Furthermore, the highest ROS level was achieved at $300\,\mu E\,m^{-2}\,s^{-1}$. In addition to the active NADPH oxidase, the limited $CO_2$ and sufficient $O_2$ under excess light were both prone to induce photorespiration that delivered electrons to $O_2$ and finally generated ROS. Consequently, mixotrophic culture with excess light was the worst condition for cells, which was in accordance with the results of $F_v/F_m$, ETR and NPQ.

**Central carbon metabolism influenced product biosynthesis.** The algal cells displayed different carbon behaviors and photosynthetic properties as light developed to excess, which meant light acted as a key factor to control the cell metabolism. Nine high-quality transcript profiles were generated with high reproducibility among the three biological replicates (Supplementary Fig. 1). As shown in Fig. 2, Kyoto Encyclopedia of Genes and Genomes (KEGG) enrichment analysis suggested heterotrophic culture had more powerful influences on ribosome biogenesis comparing with mixotrophic culture. It upregulated the biosynthesis of 90S pre-ribosome components, which might determine the different cellular morphologies under the various culture modes (Supplementary Fig. 2). Subsequently, the differential genes involved in amino acid (12 numbers), carbohydrate (10 numbers) and lipid (9 numbers) metabolism were screened. Heterotrophic culture had a great capacity to promote the absorption of extracellular nitrate by upregulating *Cz08g30210*, and the performance of assimilatory nitrate reduction was enhanced by potentially accelerating amino acid metabolism. Moreover, *Cz13g11170* and *Cz04g15230* regulating fructose and mannose metabolism from fructose-6-phosphate were downregulated. Since results of carbon partitioning revealed *C. zofingiensis* under heterotrophic culture tended to guide carbon flux from protein into lipid directly, the changing genes might then be in the charge of this carbon behavior. However, heterotrophic culture had less influence on the photosynthetic capacity (Supplementary Fig. 2). In addition, excess light limited the photosynthetic efficiency by regulating large amounts of genes associated with photon capture, photosynthetic electron transport and carbon fixation (Supplementary Fig. 3). Almost all genes involved in light capture and electron transport were downregulated under excess light. However, capacity of lipid biosynthesis was enhanced by upregulating *Cz04g05080*, which was beneficial for the accumulation of hexadecenoic acid, octadecanoic acid and octadecenoic acid. In addition, as darkness and excess light upregulated *Cz01g40120* that promoted pathway from pyruvate to phosphoenolpyruvate, they had the potential to enhance the carbon availability for product biosynthesis. The Goatools (GO) was then applied to summarize the functional genes. Comparing with light intensity at $50\,\mu E\,m^{-2}\,s^{-1}$, darkness and excess light both regulated genes related to cell death and apoptotic process, suggesting they influenced the cell cycle progression. The previous study stated that the programmed cell death involved $H_2O_2$ in various ways and excess of $H_2O_2$ led to changes in gene expression and cell death[31]. Therefore, excess light exhibited more significant impacts to downregulate the relative genes, which was mainly ascribed to the highest level of ROS. In addition, regulation of Wnt signaling pathway was beneficial for cell division and survival, and its downregulation in *Cz13g05130*, *Cz17g13110*, and *Cz07g27090* at $0\,\mu E\,m^{-2}\,s^{-1}$ revealed that heterotrophic culture induced cell growth, whereas limited its division for biomass accumulation. As phospholipids could be transferred to diacylglycerol for triacylglycerol synthesis, the downregulation of *Cz03g23260*, *Cz03g23240*, and *Cz03g23250* at both 0 and

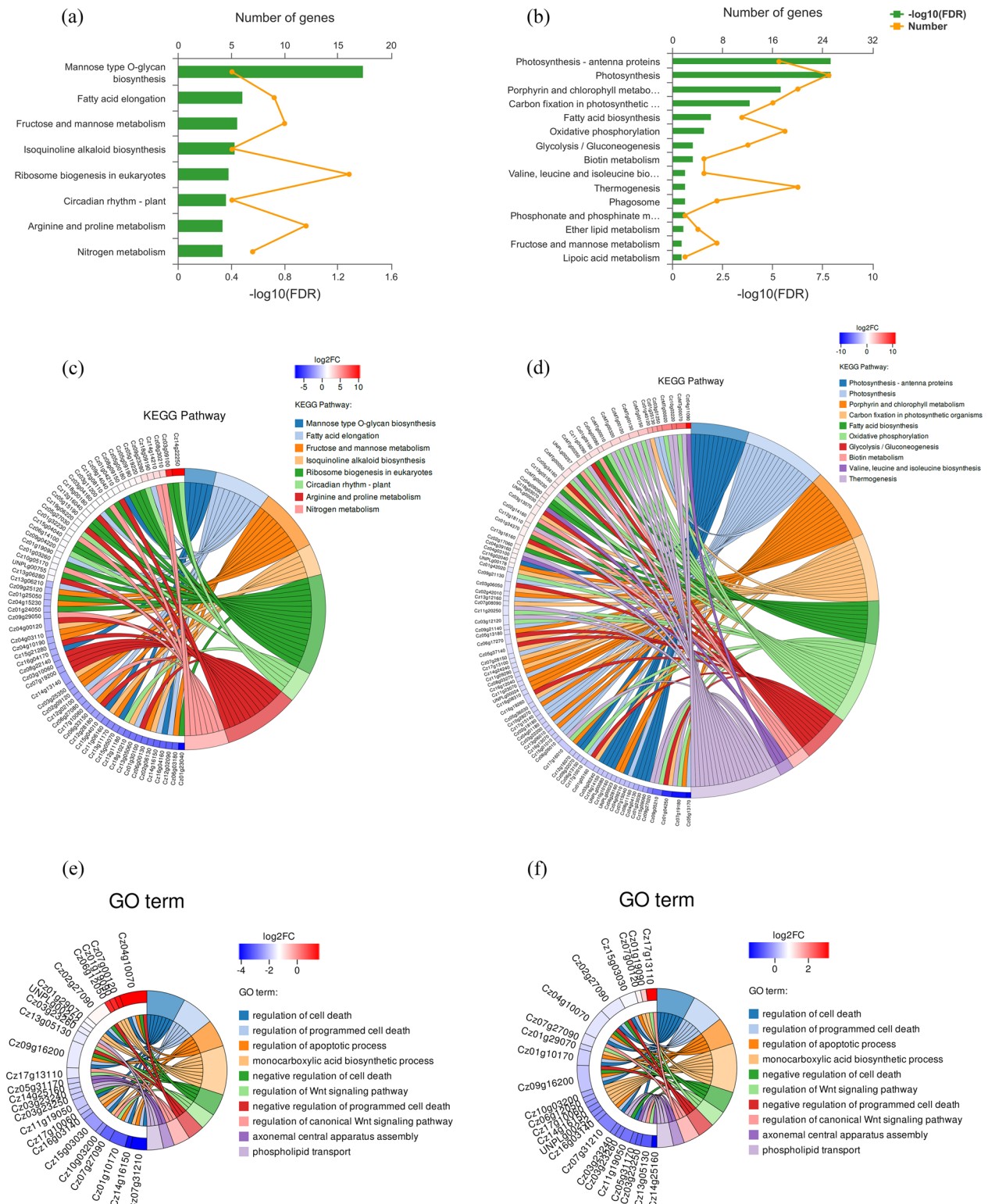

**Fig. 2 Comparison of transcriptome analysis of *C. zofingiensis* comparing under $0\,\mu E\,m^{-2}\,s^{-1}$, $50\,\mu E\,m^{-2}\,s^{-1}$ and $300\,\mu E\,m^{-2}\,s^{-1}$.** KEGG enrichment analysis by comparing $50\,\mu E\,m^{-2}\,s^{-1}$ with $0\,\mu E\,m^{-2}\,s^{-1}$ (**a**), and $50\,\mu E\,m^{-2}\,s^{-1}$ with $300\,\mu E\,m^{-2}\,s^{-1}$ (**b**). Differentially expressed genes enriched in KEGG by comparing $50\,\mu E\,m^{-2}\,s^{-1}$ with $0\,\mu E\,m^{-2}\,s^{-1}$ (**c**), and $50\,\mu E\,m^{-2}\,s^{-1}$ with $300\,\mu E\,m^{-2}\,s^{-1}$ (**d**). Go enrichment analysis by comparing $50\,\mu E\,m^{-2}\,s^{-1}$ with $0\,\mu E\,m^{-2}\,s^{-1}$ (**e**), and $50\,\mu E\,m^{-2}\,s^{-1}$ with $300\,\mu E\,m^{-2}\,s^{-1}$ (**f**). $N = 3$.

$300\,\mu E\,m^{-2}\,s^{-1}$ suggested the triacylglycerol synthetic metabolism was suppressed to some extent.

For detailed description of cell behavior and metabolism, central carbon metabolism was then analyzed to reveal the relationship between cell behavior and path rate of carbon flux map under different light intensities (Supplementary Fig. 4). As shown in Table 2, the path rates to carotenoid and lipid were both enhanced under excess light. Combining with the highest $D_P$, the lowest $\mu_{glucose}$ and Rubisco activity, excess light had a greater potential to distribute intracellular carbon molecules into valuable

**Table 2 Path rates of precursors for carotenoid and lipid, as well as rate of fatty acid under different light intensities[a].**

| Path rate (×10⁻¹) | 0 µEm⁻²s⁻¹ | 50 µEm⁻²s⁻¹ | 300 µEm⁻²s⁻¹ |
|---|---|---|---|
| Carotenoid precursor | 0.24 ± 0.01 | 0.39 ± 0.02 | 0.58 ± 0.03 |
| Lipid precursor | 15.62 ± 0.78 | 12.05 ± 0.60 | 14.97 ± 0.75 |
| C16:0 | 5.34 ± 0.27 | 3.91 ± 0.20 | 3.98 ± 0.20 |
| C16:1 | 1.91 ± 0.10 | 1.45 ± 0.07 | 1.12 ± 0.06 |
| C16:2 | 1.64 ± 0.08 | 1.36 ± 0.07 | 1.02 ± 0.05 |
| C16:3 | 0.34 ± 0.02 | 0.79 ± 0.04 | 0.72 ± 0.04 |
| C16:4 | 0.03 ± 0.00 | 0.24 ± 0.01 | 0.27 ± 0.01 |
| C18:0 | 10.20 ± 0.51 | 8.02 ± 0.40 | 10.73 ± 0.54 |
| C18:1 | 10.01 ± 0.50 | 7.36 ± 0.37 | 10.15 ± 0.51 |
| C18:2 | 4.51 ± 0.23 | 3.95 ± 0.20 | 4.15 ± 0.21 |
| C18:3 | 1.36 ± 0.07 | 2.02 ± 0.10 | 2.02 ± 0.10 |
| C18:4 | 0.06 ± 0.00 | 0.13 ± 0.01 | 0.13 ± 0.01 |

[a]Values are means ± SD, $N = 3$.

products from protein. Although darkness triggered a higher path rate to lipid than the most suitable light intensity, it restricted the path rate to carotenoid, which was in consistent with the changes of pigment content and composition (Fig. 1h, i). Considering the highest $\mu_{glucose}$ and lowest $\alpha_C$, darkness paved the way to lipid biosynthesis by rapidly converting glucose to protein and then into lipid. Moreover, light exhibited more powerful capacity to induce PUFAs accumulation, for path rates to C16:3, C16:4, C18:3, and C18:4 under light were higher than those in dark. As path rates to C16:0 and C18:1 in dark were higher than those under suitable light intensity, heterotrophic culture is of great significance for algal-derived biofuel production, due to its advantages in higher lipid content and better fatty acid profile. Energy source and consumption were different under the various treatments. As shown in Fig. 3a, ATP for biosynthesis largely increased under excess light, which was mainly attributed to the reasons that lipid biosynthesis required more energy than protein and carbohydrate at the same carbon molecules, and algal cells tended to save energy under stress conditions[4]. As the glucose uptake rate was limited under excess light, the NADPH from pentose phosphate (PP) pathway was then decreased, whereas its use for lipid biosynthesis was increased (Fig. 3c). Therefore, algal cell chose lipid biosynthesis as a valid approach to pull through stress conditions. In addition, NADPH used for lipid biosynthesis was also enhanced under heterotrophic culture than that under suitable light, in agreement with the change of lipid content. Since excess light boosted cell respiration with more ATP molecules from NADH ($p = 0.045$) and FADH₂ ($p = 0.005$), it might then result in the highest ROS level (Fig. 3b). The most suitable light had less influence on cell respiration, which could get a higher biomass concentration in combination with the energy from photosynthesis (Fig. 1e, j).

Anaplerotic reaction in cell can balance Embden-Meyerhof pathway (EMP), PP pathway, and TCA cycle[32]. The fraction of oxaloacetate was enhanced greatly from phosphoenolpyruvate, while decreased largely from malate under excess light (Fig. 3d). As clockwise TCA cycle performs to product energy, whereas the reverse is capable of providing intermediate metabolites for biosynthesis[33], fraction changes of oxaloacetate revealed that excess light boosted carbon availability by promoting the reverse cycle. Accordingly, fraction of pyruvate from malate and fraction of phosphoenolpyruvate from oxaloacetate were tremendously increased under excess light. Since pyruvate and phosphoenolpyruvate were the central intermediate metabolites linking EMP, PP pathway, TCA cycle, and biosynthesis of lipid and carotenoid, excess light potentially increased the carbon availability by reducing the rigidity of central carbon metabolism. Oppositely,

heterotrophic culture strengthened performances of glycolysis and TCA cycle as a means to increase the carbon availability[33].

**High C/N ratio boosted productivity of high-value products.** Although carbon availability can be increased under stress conditions, the strategy to ensure its rapid consumption may be of great significance for the continuous cell progression. Therefore, nitrogen-repletion fed-batch culture (NRFC) and nitrogen-starvation fed-batch culture (NSFC) to regulate cellular C/N balance were applied to create two physiological steady-growth states of algal cells, one for green growth stage and the other for lipid biosynthesis. As shown in Fig. 4, NRFC and NSFC increased biomass concentration with no significant difference under heterotrophic culture ($p = 1.000$), which might be due to the low requirement for protein biosynthesis. At 50 µE m⁻² s⁻¹, NRFC significantly heightened biomass concentration more than NSFC ($p < 0.001$). Our previous study revealed that nitrogen starvation limited the photosynthetic efficiency and nutrient absorption, leading to a lower cell growth rate than those under nitrogen repletion, and thus the complete nutrients were profitable for the green growth[18]. Interestingly, at 300 µE m⁻² s⁻¹, NSFC largely improved biomass concentration with one-fold increase than NRFC at 300 µE m⁻² s⁻¹, which was close to that obtained by NRFC at 50 µE m⁻² s⁻¹ ($p = 0.211$). The enhanced biomass accumulation might be ascribed to the active metabolic networks under nitrogen starvation, which reversely induced more carbon molecules absorbed into cells. It could be concluded that nitrogen repletion worked against rapid cell growth under the stress conditions, which then broke the routine that setting suitable environment was the best choice for the highest cell activity and biomass in microalgal production.

Moreover, NSFC at 300 µE m⁻² s⁻¹ increased the content and productivity of high-value products (Fig. 4 and Table 3). As shown in Fig. 4c–f, although excess light promoted carotenoid proportion in pigment, it significantly reduced pigment contents comparing with those at 50 µE m⁻² s⁻¹ ($p < 0.05$). However, NSFC at 300 µE m⁻² s⁻¹ resulted in a higher carotenoid content than NRFC ($p = 0.044$), which was close to that under a suitable condition ($p = 0.175$). The fed-batch cultures also influenced fatty acid profiles. Considering the stability and caloricity of oil quality, contents of C16:0 and C18:1 were the important indicators for biodiesel[34]. As shown in Fig. 4g, h, heterotrophic culture increased biodiesel performance under NSFC, and made no difference under NRFC. Notably, although excess light reduced biodiesel performance under NSFC, it enhanced the performance under NRFC. Therefore, cellular C/N ratio that could construct different intracellular microenvironment was an important index for algal-derived biodiesel production. Similarly, difference C/N ratios led to the different effects of darkness and excess light on monounsaturated fatty acid (MUFA) and PUFA (Fig. 4i, j). Darkness and excess light both acted opposite effects on MUFA. Excess light significantly decreased PUFA content ($p = 0.004$) under NRFC, while showed no effect ($p = 0.109$) under NSFC. In addition, NRFC significantly increased astaxanthin content ($p < 0.001$) and decreased lutein content ($p < 0.001$) at 300 µE m⁻² s⁻¹ (Fig. 4k). Since combination of excess light and NSFC enhanced biomass, lipid and astaxanthin accumulation, it brought out the highest productivity of fatty acids and astaxanthin. Comparing with the productivity under suitable conditions, C18:2 was improved by 63.20% and other PUFAs were increased at least one fold, as well as astaxanthin realized a 2.49-fold increase (Table 3).

**Key roles of carbon availability and path rate for product.** The outstanding advantage of NSFC was that it could achieve a

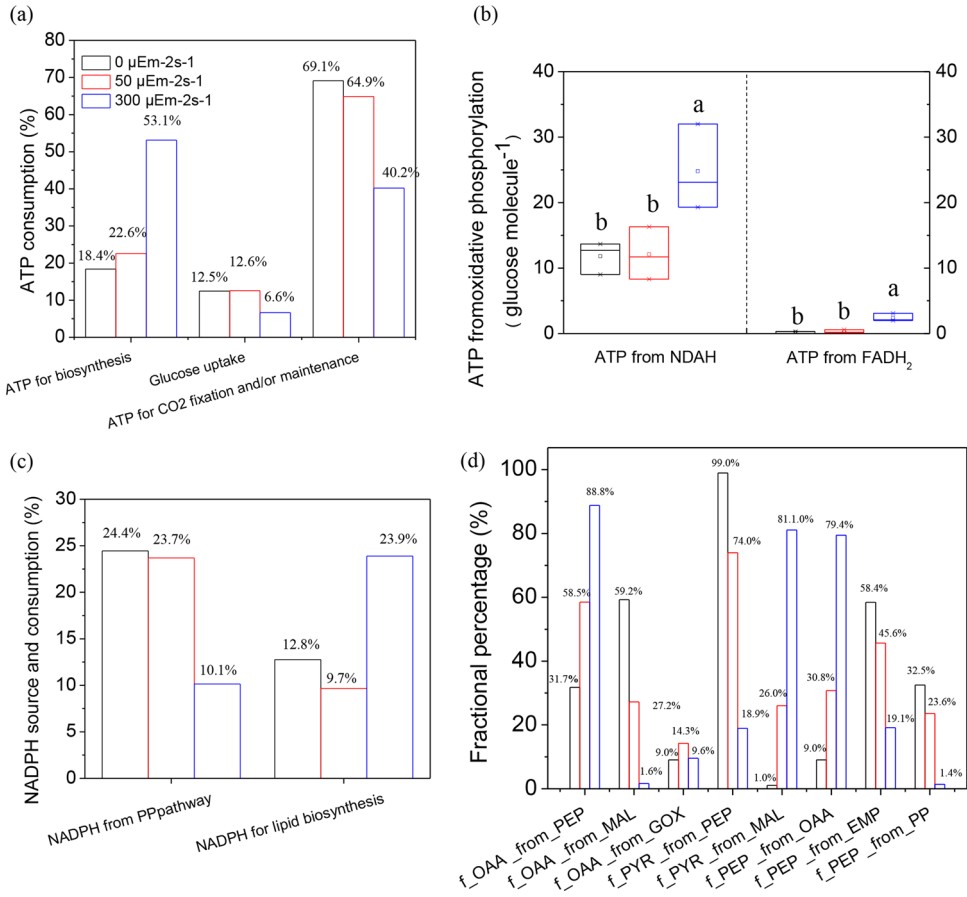

**Fig. 3 Effect of light intensity on cofactors and anaplerotic reaction.** Changes of cofactor conditions as ATP consumption (**a**), ATP source (**b**), NADPH source and consumption (**c**), and anaplerotic reaction (**d**) under different light intensities. Post-hoc comparison, different superscript letters indicate significant difference ($p < 0.05$), $N = 3$.

comparable biomass with that under suitable environment and the highest productivity of the most valuable products. This interesting phenomenon might be related to the accessible carbon and nitrogen molecules. As shown in Fig. 5, NSFC could boost the acetyl-CoA and pyruvate contents under different light intensities. Since acetyl-CoA and pyruvate were the central intermediate metabolites in carbon metabolism, their improvement suggested that intracellular carbon availability was enhanced. Acetyl-CoA was the main precursor for lipid synthesis and could be oxidized in the TCA cycle to produce NADH, $FADH_2$, ATP, and $CO_2$. Pyruvate molecules was the central metabolite in carbon-alternative pathways as it could be orientated into carotenoid, alanine, leucine, and valine, and other amino acids through TCA cycle and oxaloacetate metabolism. The sufficient extracellular carbon molecules and active anaplerotic reaction further contributed to this enhancement, and it could be concluded as the result of disturbing the former C/N balance. Nitrogen starvation guaranteed the pathways stay active for biosynthesis of lipid and astaxanthin. Their productivities, therefore, could be largely increased as long as intracellular carbon molecules were sufficient. However, normally applied stress conditions, such as excess light and nitrogen starvation, provided carbon molecules by promoting the degradation of intracellular proteins. As cultivation time went on, the cellular morphology finally kept at a relatively steady level with a certain fixed biomass composition[4]. As could be seen from Fig. 5b, g, h, NSFC, with a higher ROS level, had no significant influence on protein content comparing with NRFC under excess light ($p = 0.128$). However, it limited carbohydrate accumulation that provided precursors and

energy for lipid biosynthesis. Therefore, the residual protein, source of carbon molecules, then performed the critical role for subsistence and would not degrade in spite of the fact biosynthetic capacity of lipid and astaxanthin had yet reached the highest. Although transcriptional level and path rate of fatty acid biosynthesis revealed that excess light accelerated lipid accumulation preferably, algal cells inherently constrained carbon molecules as residual protein substances to guarantee their survival advantages. The improved carbon availability induced by NSFC under excess light was guided into lipid directly, not through the transfer station of carbohydrate, which was more efficient for glucose utilization. Therefore, feeding carbon source under stress conditions could emphatically provide a rapid direction for converting nutrients into high-value products, which then had less influences on biomass accumulation.

Another benefit of NSFC was it constructed a stressing environment through nitrogen starvation, which made it possible for the rapid biosynthesis of lipid and astaxanthin, as well as for continuous cell cycle progression. As shown in Fig. 5j, darkness and excess light promoted lycopene accumulation, the main precursor for carotenoid biosynthesis[35]. However, its content under NSFC was rarely detected, which might indicate a fast carbon-conversion rate in carotenoid biosynthesis. As α-carotene was the precursor for biosynthesis of lutein, whose content was the highest at $50\,\mu E\,m^{-2}\,s^{-1}$, its decrease under darkness and excess light suggested these conditions were beneficial for biosynthesis of secondary metabolites[35]. In addition, NSFC significantly decreased lutein content at $0\,\mu E\,m^{-2}\,s^{-1}$ ($p = 0.009$) and $50\,\mu E\,m^{-2}\,s^{-1}$ ($p = 0.021$), while it had no significant influence under excess light

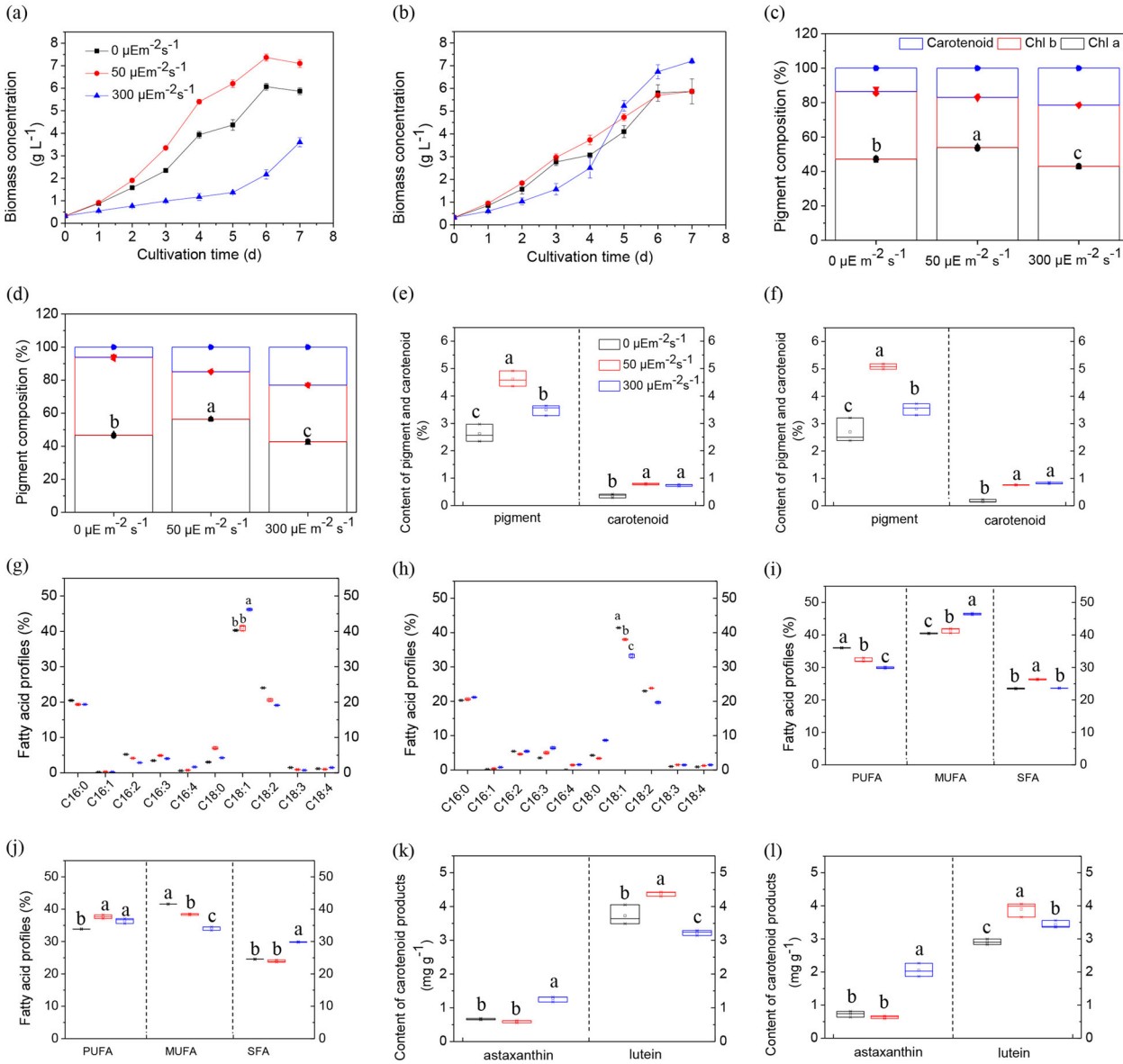

**Fig. 4 Effect of NRFC and NSFC on cell behavior.** Growth profiles of *C. zofingiensis* under at NRFC (**a**) and NSFC (**b**). Pigment composition at NRFC (**c**) and NSFC (**d**), as well as pigment and carotenoid contents at NRFC (**e**) and NSFC (**f**). Fatty acid profile at NRFC (**g**) and NSFC (**h**), fatty acid saturability at NRFC (**i**) and NSFC (**j**), as well as astaxanthin and lutein content at NRFC (**k**) and NSFC (**l**). The metabolites: Chl *a* chlorophyll *a*, Chl *b* chlorophyll *b*, PUFA polyunsaturated fatty acid, MUFA monounsaturated fatty acid, SFA saturated fatty acid. Post-hoc comparison, different superscript letters indicate significant difference ($p < 0.05$), $N = 3$.

($p = 0.058$). However, unlike the lutein biosynthesis, the lower precursor β-carotene resulted in a higher astaxanthin content, which revealed that biosynthesis of lutein was in a carbon-storage manner and astaxanthin as a carbon-usage form.

## Discussion

It is now universally acknowledged that valuable product accumulation in microalgae cannot go without efficient carbon conversion among the main carbon sink[36]. Harnessing cultivation condition and environment is considered as a useful approach to produce the products purposefully. However, by assessment of biomass component in a black box, accuracy and validity of the approach are challenged for the lack of comprehensive understanding of carbon behaviors[37]. The same treatments will then exhibit different influences on biomass and product accumulation.

As could be obtained from this study, carbon availability and path rate of central carbon metabolism were two fundamental factors in microalgal production. The protein degradation through TCA cycle provided carbon molecules and energy for biosynthesis of lipid and carotenoid. The path rates of α-Ketoglutarate to oxaloacetate from succinate, fumarate and malate apparently increased under excess light contrasting with $50\,\mu E\,m^{-2}\,s^{-1}$, ascribed to the insufficient biosynthesis of protein or its degradation[38]. Then, results from kinetic model, transcriptome analysis and $^{13}$C-MFA suggested there was in dire need of the reservation of carbohydrate and lipid to confirm the survival advantages, and elevated carbon molecules were guided into these carbon sinks. However, darkness downregulated *Cz16g00050*, *Cz02g14160*, *Cz13g05150*, *Cz04g09090*, *UNPLg00257*, and *Cz01g09160* that participated into lipid synthesis, which demonstrated that the increased lipid content was attributed to the elevated carbon availability (Supplementary Fig. 5). Similarly, although excess light

**Table 3 Productivity of high-value products of C. zofingiensis[a].**

| Products | NRFC (mg L$^{-1}$ h$^{-1}$) | | | NSFC (mg L$^{-1}$ h$^{-1}$) | | |
|---|---|---|---|---|---|---|
| | 0 μE m$^{-2}$ s$^{-1}$ | 50 μE m$^{-2}$ s$^{-1}$ | 300 μE m$^{-2}$ s$^{-1}$ | 0 μE m$^{-2}$ s$^{-1}$ | 50 μE m$^{-2}$ s$^{-1}$ | 300 μE m$^{-2}$ s$^{-1}$ |
| C16:0 | 2.45 ± 0.09 | 2.17 ± 0.20 | 1.56 ± 0.11 | 2.81 ± 0.62 | 2.20 ± 0.13 | 4.03 ± 0.03 |
| C16:1 | 0.02 ± 0.00 | 0.03 ± 0.00 | 0.02 ± 0.00 | 0.02 ± 0.00 | 0.04 ± 0.00 | 0.15 ± 0.02 |
| C16:2 | 0.62 ± 0.02 | 0.47 ± 0.04 | 0.23 ± 0.01 | 0.75 ± 0.16 | 0.50 ± 0.06 | 1.03 ± 0.04 |
| C16:3 | 0.41 ± 0.02 | 0.55 ± 0.06 | 0.32 ± 0.02 | 0.49 ± 0.10 | 0.54 ± 0.09 | 1.22 ± 0.06 |
| C16:4 | 0.08 ± 0.00 | 0.08 ± 0.00 | 0.13 ± 0.00 | 0.00 | 0.15 ± 0.03 | 0.30 ± 0.01 |
| C18:0 | 0.36 ± 0.00 | 0.78 ± 0.10 | 0.34 ± 0.03 | 0.59 ± 0.12 | 0.36 ± 0.01 | 1.65 ± 0.06 |
| C18:1 | 4.79 ± 0.16 | 4.62 ± 0.48 | 3.73 ± 0.30 | 5.74 ± 1.23 | 4.07 ± 0.28 | 6.32 ± 0.21 |
| C18:2 | 2.86 ± 0.09 | 2.31 ± 0.24 | 1.54 ± 0.10 | 3.18 ± 0.68 | 2.55 ± 0.19 | 3.77 ± 0.07 |
| C18:3 | 0.18 ± 0.01 | 0.10 ± 0.01 | 0.06 ± 0.00 | 0.14 ± 0.03 | 0.16 ± 0.01 | 0.28 ± 0.02 |
| C18:4 | 0.14 ± 0.01 | 0.11 ± 0.01 | 0.12 ± 0.00 | 0.13 ± 0.03 | 0.13 ± 0.02 | 0.28 ± 0.01 |
| Lutein (×10$^{-1}$) | 1.30 ± 0.07 | 1.85 ± 0.08 | 0.69 ± 0.04 | 1.01 ± 0.02 | 1.36 ± 0.07 | 1.46 ± 0.07 |
| Astaxanthin (×10$^{-2}$) | 2.30 ± 0.01 | 2.52 ± 0.19 | 2.72 ± 0.28 | 2.56 ± 0.49 | 2.22 ± 0.12 | 8.80 ± 0.79 |

[a]Values are means ± SD, N = 3.

downregulated *Cz02g32280* involving conversion of lycopene from phytoene (Supplementary Fig. 6), it increased lycopene and astaxanthin content. Consequently, carbon availability exhibited a higher potential for controlling product biosynthesis. However, once the protein content reduced to threshold value for cell survival, intracellular carbon availability was the major limiting factor for product biosynthesis and cell reproduction. The C/N ratio was increased under this circumstance. In this situation, algal cell stopped to grow and maintained necessary substances and energy for survival, even though they were capable of the next cell cycle progression (Fig. 6). To overcome the insufficient source of carbon molecules and energy for cell metabolism, NSFC could simultaneously realize sufficient carbon molecules and high path rates for biosynthesis of lipid and astaxanthin. Although the C/N ratio was further increased, NSFC tremendously increased the biomass concentration and contents of lipid and astaxanthin. The increase of acetyl-CoA and pyruvate, without any side effects on protein content, potentially expedited the utilization of the available carbon molecules, as well as affected fatty acid and carotenoid biosynthesis. Ultimately, the algal cell tended to regrowth in forms of biomass accumulation (Fig. 6).

One important point needs to be considered when providing extracellular carbon molecules was the ROS level. ROS were reported to be favorable for lipid and astaxanthin, while against for cell growth[18]. To increase the ROS level, heterotrophic culture induced the NADPH oxidase to be more active, and mixotrophic culture under excess light improved photorespiration and NADPH oxidase activity. Numerous studies stated algal cells were able to recover from nutrient starvation to green vegetative cell[39]. The recovery reduced the ROS level and downregulated the path rates of lipid and astaxanthin. Oppositely, strategies to enhance carbon availability whereas decrease nitrogen availability further promoted lipid and astaxanthin accumulation with the highest ROS level. Our study suggested although algal cells reached a high ROS level, the sufficient carbon molecules and energy could impel the cell progression. Therefore, feeding extra carbon source under excess light was attractive in microalgae cultivation, especially for outdoor cultivation. The high sunlight intensity could be utilized efficiently for a higher productivity of the high-value products.

Normally, stress-based strategies were capable of enhancing the biomass value by upregulating path rates of lipid and several carotenoids, whereas largely reacting against cell growth or even inducing cell death. Since the strategies elevated available carbon molecules through conversion of the main carbon sinks[26], they greatly damaged the cellular environmental homeostasis. Once protein content achieved threshold value for survival, process of

product accumulation would be interrupted and the disturbed environmental homeostasis could no longer support the next cell progression. The intracellular carbon availability was then the limiting factor for microalgal production under such stress-based strategies. Therefore, harnessing C/N balance in microalgal production possessed the following three meanings: (1) guarantee a suitable condition for rapid cell growth; (2) construct a stressing environment for product biosynthesis, and (3) provide enough carbon availability and high ROS level for both rapid cell growth and product biosynthesis. Regulating cellular C/N ratio was critical in microalgal production as it could realize a higher lipid and astaxanthin contents without affecting cell growth. This study then overcame the biggest conflict between biomass concentration and accumulation of high-value products in microalgal production.

## Methods

**Cultures and growth.** *Chromochloris zofingiensis* (ATCC 30412) was obtained from the American Type Culture Collection (ATCC, Rockville, MD, USA). The strain was cultured in BG-11 media. The cultures were incubated in 250-mL Erlenmeyer flasks containing 100 mL Kuhl media at 25 °C for 4 days and illuminated with a continuous light intensity of 30 μE m$^{-2}$ s$^{-1}$. *C. zofingiensis* grows well in Kuhl medium, which consists of (per L): KNO$_3$ 2.02 g, NaH$_2$PO$_4$·2H$_2$O 0.7 g, Na$_2$HPO$_4$·12H$_2$O 0.181 g, MgSO$_4$·7H$_2$O 0.247 g, CaCl$_2$·2H$_2$O 14.7 mg, FeSO$_4$·7H$_2$O 6.95 mg, H$_3$BO$_3$ 0.061 mg, MnSO$_4$·H$_2$O 0.169 mg, ZnSO$_4$·7H$_2$O 0.287 mg, CuSO$_4$·5H$_2$O 0.0025 mg, (NH$_4$)$_6$M$_{O7}$O$_{24}$·4H$_2$O 0.01235 mg. The pH value was adjusted to 6.5 prior to autoclaving.

The seed cells were grown in Kuhl with 5 g L$^{-1}$ glucose as the carbon source. The cells were cultivated in 250-mL Erlenmeyer flasks containing 100 mL of growth media at 23 °C under continuous illumination of 0, 30, 50, 150, and 300 μE m$^{-2}$ s$^{-1}$. During fed-batch culture, the feeding media and conditions were set according to our previous study[40]. For metabolic flux analysis of the labeled group, 20 and 100% U-13C glucoses were set as the carbon source and 1 mL seed cells were washed and inoculated in 100 mL labeled media for cell growth. The other conditions were same as the unlabeled.

## Model development

*Model development of carbon partitioning.* The model was constructed to predict storage carbon behaviors based on the following assumptions: (1) the cultivation conditions were constant, (2) the extracellular metabolites during the exponential growth phase were seldom and had no inhibitory effect on microalgae, (3) nitrogen was the only limited factor for carbon partitioning, and (4) carbohydrate and lipid concentrations were proportional to the biomass concentration. The behaviors of the carbon sinks (protein, carbohydrate, and lipid) were initially designed based on the general physiological change of microalgae. The nutrients in the media were absorbed and assimilated into protein and carbohydrate firstly. During the growth stage, a part of protein was degraded and converted into carbohydrate, lipid and other functional compartment. Partial carbohydrate was reallocated into lipid, especially as the environmental conditions got worse. Therefore, the global carbon flux can then be simplified into two specific fluxes, which lead to the final

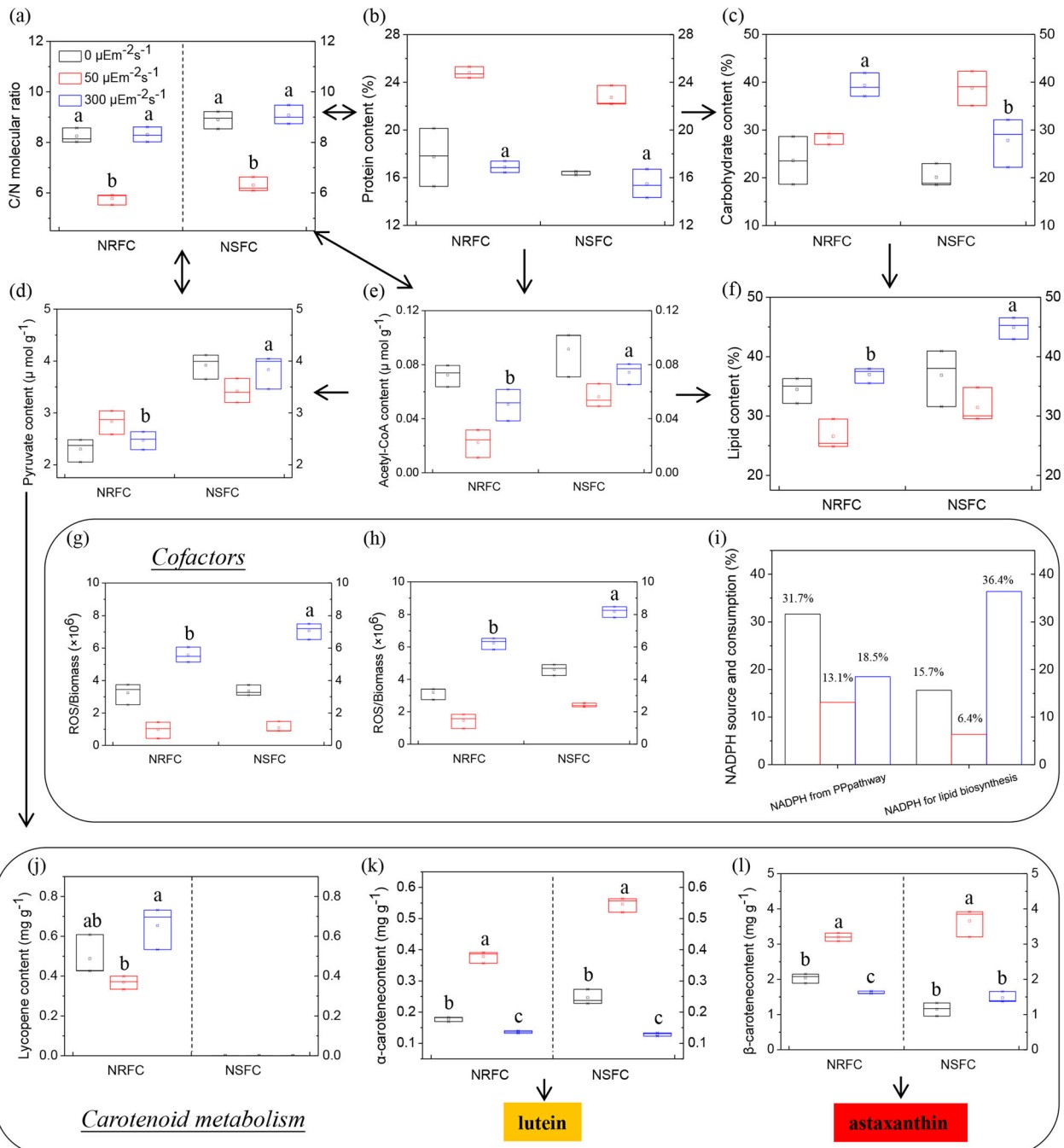

**Fig. 5 C/N balance regulates central metabolites and carbon partitioning, carotenoid synthetic pathway and cofactors.** C/N molecular ratio (**a**) and changes of protein (**b**), carbohydrate (**c**) and lipid (**f**) at NRFC and NSFC. Pyruvate (**d**), and acetyl-CoA (**e**) content at NRFC and NSFC. Cofactors of ROS level after 4 days (**g**) and 7 days (**h**), as well as the NADPH source and consumption (**i**) at NRFC and NSFC. Carotenoid product of lycopene (**j**), α-carotene (**k**), and β-carotene (**l**) content at NRFC and NSFC. Post-hoc comparison, different superscript letters indicate significant difference ($p < 0.05$), $N = 3$.

production of lipid. The model was established as the following equations:

$$\frac{dQ_P}{dt} = Y_{PS}\frac{dS}{dt} - D_P Q_P \tag{1}$$

$$\frac{dQ_C}{dt} = Y_{CS}\frac{dS}{dt} - \alpha_C Q_C \tag{2}$$

$$\frac{dQ_L}{dt} = Y_{CL}\alpha_C Q_C + Y_{PL}D_P Q_P \tag{3}$$

where $Q_P$, $Q_C$, and $Q_L$ were the content of protein, carbohydrate and lipid respectively (g L$^{-1}$), $Y_{PX}$ and $Y_{CX}$ represented the protein and carbohydrate yields on biomass concentration respectively (g g$^{-1}$), $Y_{CL}$ and $Y_{PL}$ represented carbohydrate and protein yields on lipid yield respectively (g g$^{-1}$), $t$ was the cultivation

time (h), $S$ was the substance concentration (g L$^{-1}$), $D_P$ and $\alpha_C$ were the conversion rate of protein and carbohydrate (h$^{-1}$) respectively. Then, the above equations could be converted into:

$$\frac{dQ_P}{dt} = Y_{PX}\mu X_0\exp(\mu t) - D_P Q_{P0}\exp(\mu_P t) \tag{4}$$

$$\frac{dQ_C}{dt} = Y_{CX}\mu X_0\exp(\mu t) - \alpha_C Q_{C0}\exp(\mu_C t) \tag{5}$$

$$\frac{dQ_L}{dt} = \frac{\mu_C}{\mu_L}\alpha_C Q_{C0}\exp(\mu_C t) + \frac{\mu_P}{\mu_L}D_P Q_{P0}\exp(\mu_P t) \tag{6}$$

where $\mu$ was the specific growth rate of (h$^{-1}$), $\mu_P$, $\mu_C$, and $\mu_L$ were the specific generating rate of protein, carbohydrate, and lipid (h$^{-1}$) respectively, $X_0$ was initial

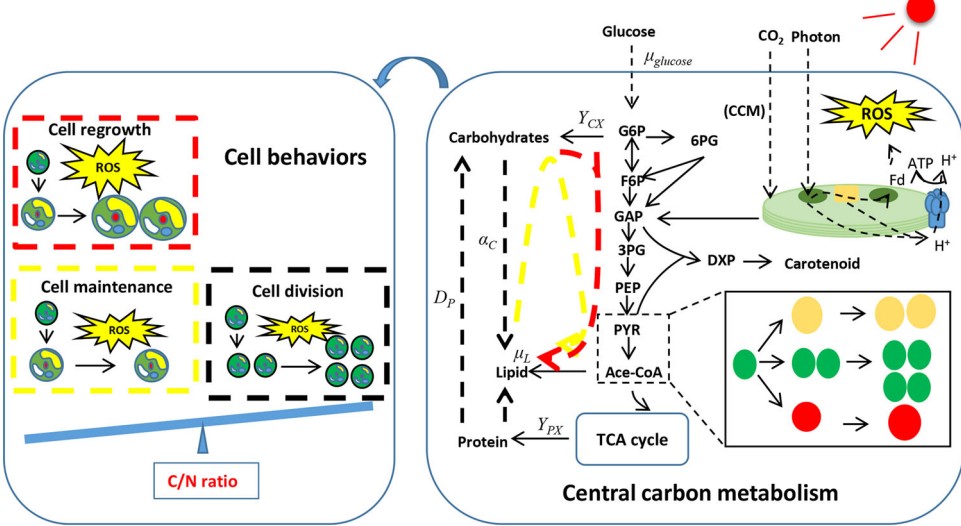

**Fig. 6 Schematic diagram of relationship between central carbon metabolism and cell behavior.** The carbon metabolites: G6P glucose-6-phosphate, F6P fructose-6-phosphate, 6PG 6-Phosphogluconic acid, GAP glyceraldehyde 3-phosphate, 3PG 3-Phosphoglycerate, PEP phosphoenolpyruvate.

cell concentration (g L$^{-1}$). The rates were influenced by protein content and could be described as:

$$\mu = \mu_{max}\left(1 - \frac{Q_{P,min}}{Q_P}\right) \qquad (7)$$

Therefore, indexes of $Y_{PX}$ and $Y_{CX}$ were also influenced by the protein content.

*Exponential fed-batch culture.* To increase biomass of *C. zofingiensis*, exponential fed-batch culture was used to supply sufficient nutrients during the exponential growth phase. The feeding nutrients during cultivation were fixed at the initial concentrations, respectively. Therefore, extracellular products were seldom produced and consumption of substance and energy was low for cell maintenance during the phase. The fed-batch culture containing nitrogen was named as nitrogen-repletion fed-batch culture (NRFC), whereas without nitrogen was defined as nitrogen-starvation fed-batch culture (NSFC). The model was expressed as the following equation:

$$\frac{dS}{dt} = F_S - \frac{1}{Y_{XS}}\frac{dX}{dt} \qquad (8)$$

where $F_S$ was the feeding concentration of substrate and $Y_{XS}$ was biomass yield on substrate. When the nutrient concentration was maintained constantly, equation was described as:

$$F_S = \frac{1}{Y_{XS}}\mu X_0 \exp(\mu t) \qquad (9)$$

Then, the volume was used as the feeding index, Eq. (9) could be described as the following equation:

$$V_{FS} = \frac{1}{Y_{XS}}\mu X_0 \exp(\mu t) V_0 / C_{FS} \qquad (10)$$

where $V_{FS}$ was the feeding rate (mL h$^{-1}$), $C_{FS}$ was the nutrient concentration in feeding media (g L$^{-1}$) and $V_0$ was the initial cultivation volume (mL).

**Analysis**

*Protein, carbohydrate, and lipid.* The lipid, carbohydrate, and protein were determined as previously described[18]. The lipid was extracted and determined by measuring the weight. The carbohydrate was extracted and determined by phenol sulfuric acid method. The protein was extracted and collected to measure the protein concentration by protein assay kit (Bio-Rad #5000002, Hercules, USA).

*Precursors for lipid and carotenoid.* For determination of acetyl-CoA content, acetyl-CoA was extracted by 1 mol L$^{-1}$ perchloric acid and measured by the acetyl-CoA assay kit (Sigma, MAK039). For determination of pyruvate content, pyruvate was extracted and measured by the pyruvate assay kit (Sigma, MAK071).

*Pigment contents.* The pigment contents were determined photometrically[41]. The cells were collected by centrifugation at 13,000 × g for 5 min. The pellet was then redissolved with 99.9% methanol and incubated at 45 °C for 24 h in dark. The extract was centrifuged at 13,000 rpm for 5 min and the supernatant was used to measure the absorbance at 470, 652.4 and 665.2 nm. The pigment concentrations were calculated according to the previous study[41].

The carotenoids were extracted by acetone and analyzed by HPLC (Waters, Milford, MA, USA) that was equipped with a Shiseido CAPCELL PAK C18 5 μm column (4.6 × 250 mm) and a 2998 photodiode array detector (Waters, Milford, MA, USA). The operational parameters were set according to the pervious study[42].

*Quantum yield of photosystem II and ROS.* The maximum quantum yield ($F_v/F_m$) of the photosystem II and electron transport rate (ETR) were measured by a pulse-amplitude-modulated fluorometer (Walz, Effeltrich, Germany) according to the previous study[18]. The ROS levels were determined with the ROS assay kit (Beyotime Institute of Biotechnology, China) as previously described[18]. The fluorescence was immediately measured by a fluorescence microplate reader (Thermo Fisher Scientific, Waltham, MA, USA).

*Enzyme activity.* The cells were collected and centrifuged for 5 min at 5000 rpm under 4 °C. Then, cell pellets were grinded under liquid nitrogen. For determination of Rubisco activity, the samples were extracted and measured according to instruction of Rubisco assay kit (Solarbio, Beijing, China). For determination of NADPH oxidase activity, the samples were extracted and measured according to kit's instruction (GENMED SCIENTIFICS INC, USA).

**RNA sequencing and differentially expressed gene analysis.** Total RNA was extracted using TRIzol reagent (Invitrogen, https://www.thermofisher.com/). The RNA quality and concentration were examined using Agilent 2100 Bioanalyzer (Agilent Technologies) and NanoDrop 2000C (Thermo Scientific). Then mRNA purification was carried out using Sera-mag Magnetic Oligo(dT) Beads (Thermo Scientific). The transcriptome libraries were prepared using the TruseqTM RNA sample pre Kit (Illumina, https://www.illumina.com) and sequenced for 2 × 150-bp runs (paired-end) using a Illumina HiSeq 4000 sequencing system (Illumina, https://www.illumina.com) by Majorbio (Shanghai) Co., Ltd, China. Reads were aligned to the *C. zofingiensis* genome[43] (https://phytozome.jgi.doe.gov/pz/portal.html#!info?alias=Org_Czofingiensis_er) with HISAT2 (https://ccb.jhu.edu/software/hisat2/index.shtml). Reads mapping to more than one location were excluded.

Gene expression was measured as the numbers of aligned reads to annotated genes using RSEM software and normalized to FPKM values. The DEGs were identified using DESeq2 software (http://biocounductor.org/packages/release/bioc/html/DESeq2.html). Genes were considered to be significantly differentially expressed if their expression values showed at least a two-fold change with an FDR adjusted $P$-value < 0.05 between control and experimental conditions. KEGG and Go enrichment analysis were carried out using KOBAS (https://kobas.cbi.pku.edu.cn/nome/do) and Goatools (https://github.com/tanghaibao/GOatools) respectively.

**$^{13}$C tracer-based metabolic flux analysis.** $^{13}$C-MFA was calculated based on the biomass component to establish the biomass reaction (Supplementary Tables S1–S3). The each nucleotide content was obtained from published literature[43]. The labeled amino acids was measured by GC-MS (7890B-5977B, Agilent). The glucose uptake was set as 100 to compare the different influences of cultures on central carbon metabolism. The abundance of labeled amino acid was modified by analyzing 100% U-$^{13}$C glucose, although the contribution of $CO_2$ to biomass was low (Supplementary Fig. 7).

**Statistics and reproducibility.** All the experiments were conducted in at least three biological replicates to ensure the reproducibility. Statistical analysis was

carried out by using SPSS software. A one-way analysis of variance (ANOVA) was used to detect the significant differences from the respective control groups for each experimental test condition.

**Reporting summary**. Further information on research design is available in the Nature Research Reporting Summary linked to this article.

## Data availability

RNA-seq data have been deposited and are available under accession number PRJNA612734. The data underlying metabolic flux analysis are provided as supplementary information. Other relevant data supporting the findings are available with supplementary materials or from the authors upon reasonable requests.

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

## Acknowledgements

This research was financially supported by Key Realm R&D Program of Guangdong Province (No. 2018B020206001) and Science and Technology Innovation Commission of Shenzhen (No. KQTD20180412181334790).

## Author contributions

H.S. and F.C. conceived, designed, and drafted the paper. H.S. and Y.R. contributed kinetic model and fed-batch culture. H.S. and H.Z. contributed metabolic flux analysis. H.S. and X.M. contributed transcriptome analysis. X.L. and Y.L. contributed measurement of photosynthetic characteristics. F.C. critically revised the manuscript.

## Competing interests

The authors declare no competing interests.
