## [Peer Review File · Communications Biology]

Reviewers' comments:

Reviewer #1 (Remarks to the Author):

My review is limited to the techno-economic presented in the paper.

The techno-economic work presented is not a techno-economic assessment. It does not include considerations for any of the capital investment of the system integrated into a cash flow system. It is limited to a simple operational cost compared to product assessment. The biomass value presented does not include considerations for the growth infrastructure.

Reviewer #2 (Remarks to the Author):

In my opinion the paper "Harnessing C/N balance to resolve the potential conflicts in economic microalgae production" is a paper that explores all the interactions in a bioprocess production using microalgae as biocatalizer. Also, the nexus between the technical and the economic approach give a panorama of how the bioprocess works in a better way with some modification balance on C/N, something that most of the paper published does not take into account in first instance.

Reviewer #3 (Remarks to the Author):

The paper "Harnessing C/N balance to resolve the potential conflicts in economic microalgae production" by Han Sun et al., present a huge work on the characterization of the carbon pathways in microalgae. The topic is addressed from many points of view, ranging from modeling, to biological observation and economic analysis. This is also the weak point of the paper, that sometimes appears not well integrated in its sections. The discussion about economics is not clearly linked to the results obtained, and it appears superficial with respect to the other sections of the paper. On the other hand, I'm not an expert in the field, but my humble opinion is that the economic assessment should be based on productivity data, that are missing in the paper. In addition, productivity per area should be considered while assessing the cost of energy from sunlight or as electricity. Maybe the authors should better describe the economic models used and the input data (especially in the case of the biomass and commercial products productivities).

However, the paper presents a sufficient novelty and many interesting data, so it may be published after major revisions.

The role of CO₂ is not well explained. In particular, experiments seem to be carried out in Erlenmeyer flasks, so that the culture is limited by CO₂. When discussing about the carbon cycle, I suggest to consider also the role of inorganic carbon.

Which is the feed of the fed batch experiments? The concentration of carbon and nitrogen is reported somewhere?

From a modeling point of view, the parameters (table 1) changed their value as effect of light intensity. So, this suggest that a dependence on light should be accounted from a modeling point of view, and the current model formulation is not appropriate. It is not clear the scope of calibrating the model and discuss the value of the parameters, that actually are not parameters, but possibly function of the light intensity. How the model is then used? Just as an observation of the experimental data? Please discuss this point.

It is not clear the meaning of "steady state" in fed batch processes. In fact, cell concentration increases (fig 4), so that a steady state assumption cannot be applied. Please explain. All the conclusions about the fed batch curves under nitrogen starvation are not clear and well discussed, and the concentration of the feed used are not discussed properly.

Other remarks:

In figure 4 a b c please use the same scale in axis

Figure 2 is hard to read. Legends are too small

Figure 6: the figure on the left is possibly misleading. Has the inclination of the balance a specific meaning? (it appears that when C/N ratio is high cell regrowth and cell maintenance is preferred)

Line 97-99: the sentence "It limited the availability of metabolic regulation, which would be designed to suppress or remove the paths with high-energy consumption." is not clear and should be re-written.

Generally a revision of English is needed.

Substitute "in a molecular level" with "at the molecular level"

Figure 1g: due to different scale, the value of F_v/F_m are not well visible

Kuhl medium is not standard, please report the composition and explain the choice

Reviewer #1 (Remarks to the Author):

My review is limited to the techno-economic presented in the paper.

The techno-economic work presented is not a techno-economic assessment. It does not include considerations for any of the capital investment of the system integrated into a cash flow system. It is limited to a simple operational cost compared to product assessment. The biomass value presented does not include considerations for the growth infrastructure.

Response: Thank you for the comment. We have transferred this section into supplementary material and changed “techno-economic analysis” to “economic estimation of operational cost and biomass value” for the accurate description. Since our economic analysis of the cost has not considered the “capital investment of the system integrated into a cash flow system”, we changed it to economic estimation of operational cost, as the reviewer suggested. Please see line 406-416 and supplementary material in red.

In addition, the biomass value was calculated based on the revenue of biomass component from specific market scenarios, which was used in previous study. We revised the expression for calculating biomass value. Please see supplementary material in red.

Reference:

1. Ruiz, J., Olivieri, G., de Vree, J., Bosma, R., Willems, P., Reith, J. H., Eppink, M. H. M., Kleinegris, D. M. M., Wijffels, R. H. & Barbosa, M. J. Towards industrial products from microalgae. *Energ. Environ. Sci.* 9, 3036-3043 (2016).
2. Sun, H., Mao, X., Wu, T., Ren, Y., Chen, F. & Liu, B. Novel insight of carotenoid and lipid biosynthesis and their roles in storage carbon metabolism in *Chlamydomonas reinhardtii*. *Bioresource Technol.* 263, 450-457 (2018).

Reviewer #2 (Remarks to the Author):

In my opinion the paper “Harnessing C/N balance to resolve the potential conflicts in economic microalgae production” is a paper that explores all the interactions in a bioprocess production using microalgae as biocatalizer. Also, the nexus between the technical and the economic approach give a panorama of how the bioprocess works in a better way with some modification balance on C/N, something that most of the paper published does not take into account in first instance.

Response: Thank you for affirming our work.

Reviewer #3 (Remarks to the Author):

The paper “Harnessing C/N balance to resolve the potential conflicts in economic microalgae production” by Han Sun et al., present a huge work on the characterization of the carbon pathways in microalgae. The topic is addressed from many points of

view, ranging from modeling, to biological observation and economic analysis. This is also the weak point of the paper, that sometimes appears not well integrated in its sections. The discussion about economics is not clearly linked to the results obtained, and it appears superficial with respect to the other sections of the paper. On the other hand, I'm not an expert in the field, but my humble opinion is that the economic assessment should be based on productivity data, that are missing in the paper. In addition, productivity per area should be considered while assessing the cost of energy from sunlight or as electricity. Maybe the authors should better describe the economic models used and the input data (especially in the case of the biomass and commercial products productivities). However, the paper presents a sufficient novelty and many interesting data, so it may be published after major revisions.

Response: Thank you for the comments. We have revised the manuscript to well integrate the sections according to the comments. The economic assessment was transferred to supplementary material and modified as “economic estimation of operational cost and biomass value” for the accurate description.

In addition, the biomass value was calculated according to previous study, which was based on the revenue of biomass component from specific market scenarios. The definition was not involved in time. Therefore, we performed the all cultivations for 7 days (Fig. 4a, b, c) and calculated the operational cost (supplementary material). Then the economic models could be better to describe the biomass and commercial products productivities.

References:

1. Ruiz, J., Olivieri, G., de Vree, J., Bosma, R., Willems, P., Reith, J. H., Eppink, M. H. M., Kleinegris, D. M. M., Wijffels, R. H. & Barbosa, M. J. Towards industrial products from microalgae. *Energ. Environ. Sci.* 9, 3036-3043 (2016).

The role of CO₂ is not well explained. In particular, experiments seem to be carried out in Erlenmeyer flasks, so that the culture is limited by CO₂. When discussing about the carbon cycle, I suggest to consider also the role of inorganic carbon.

Response: Thank you for the comment. We supplemented the experiment of Rubisco activity' measurement and then explained the role of CO₂ in carbon cycle. Please see line 165-173, 231, 552-558 and Fig. 1 in red.

Which is the feed of the fed batch experiments? The concentration of carbon and nitrogen is reported somewhere?

Response: Thank you for the comment. We have added the information. The feeding media and conditions were according to our pervious study. Please see line 446-448 in red.

From a modeling point of view, the parameters (table 1) changed their value as effect of light intensity. So, this suggest that a dependence on light should be accounted from a modeling point of view, and the current model formulation is not appropriate.

It is not clear the scope of calibrating the model and discuss the value of the

parameters, that actually are not parameters, but possibly function of the light intensity. How the model is then used? Just as an observation of the experimental data? Please discuss this point.

Response: Thank you for the comment. We have added the explanations to use this model in the manuscript. Design of the model aimed at exploring the changes of carbon partitioning and glucose uptake under different light intensities. Its combination with Rubisco activity and metabolic flux analysis could map the carbon flux from carbon source to protein, carbohydrate, lipid and pigment. The elevated carbon availability for product biosynthesis could be from the extracellular carbon source and conversion of carbon partitioning. Their combinations could distinguish the influences. Please see line 149-150, 231-233 and 236-237 in red.

It is not clear the meaning of “steady state” in fed batch processes. In fact, cell concentration increases (fig 4), so that a steady state assumption cannot be applied. Please explain. All the conclusions about the fed batch curves under nitrogen starvation are not clear and well discussed, and the concentration of the feed used are not discussed properly.

Response: Thank you for the comment. We have changed “steady state” into “steady-growth state”. Please see line 84 and 275 in red.

In addition, we have added the description and discussion of the fed batch curves under nitrogen starvation. The N-starvation guaranteed active pathways for biosynthesis of lipid and astaxanthin, and limited photosynthetic efficiency and

nutrient absorption. Please see line 279-282, 285-288, 327-329 and 344-346 in red.

The model of exponential fed-batch culture aims at controlling the nutrients at a fixed concentration by calculating the nutrient consumption per hours. Theoretically, the feeding nutrients during cultivation were fixed at the initial concentrations, respectively. We have added the explanation. Please see line 488-489 in red.

Other remarks:

In figure 4 a b c please use the same scale in axis

Response: Thank you for the comment. We have modified the scale. Please see Fig. 4.

Figure 2 is hard to read. Legends are too small

Response: Thank you for the comment. We have enlarged the legends and improved the resolution. Please see Fig. 2.

Figure 6: the figure on the left is possibly misleading. Has the inclination of the balance a specific meaning? (it appears that when C/N ratio is high cell regrowth and cell maintenance is preferred)

Response: Thank you for the comment. We have added the explanation. Once the protein content decreased to its threshold value for cell survival, the C/N ratio was increased. Algal cell stopped to grow and maintained substance and energy for survival. Although the C/N ratio was further improved under NSFC, NSFC tremendously increased the biomass concentration and contents of lipid and

astaxanthin. The algal cell tended to regrowth in forms of biomass accumulation.

Plases see line 386-388, 390-393 and 395-396 in red.

Line 97-99: the sentence “It limited the availability of metabolic regulation, which would be designed to suppress or remove the paths with high-energy consumption.” is not clear and should be re-written.

Response: Thank you for the comment. We have re-written the sentence. Please see line 99-100 in red.

Generally a revision of English is needed.

Response: Thank you for the comment. The English was revised. Please see the manuscript.

Substitute “in a molecular level” with “at the molecular level”

Response: Thank you for the comment. We have changed it. Please see line 95 and 111 in red.

Figure 1g: due to different scale, the value of F_v/F_m are not well visible

Response: Thank you for the comment. We have changed it. Please see Fig. 1.

Kuhl medium is not standard, please report the composition and explain the choice

Response: Thank you for the comment. We have added the composition and reason.

Please see line 439-443 in red.

Reviewers' comments:

Reviewer #1 (Remarks to the Author):

Unless the authors include growth infrastructure and do a discounted cash flow rate of return analysis the economics in the paper are not of value. Statements like "Our findings provide a new orientation to maximize biomass productivity and value that makes microalgal production economically viable in various applications." Are not accurate. It is a falsity as the economic viability cannot be assessed strictly by understanding the value of the biomass. Microalgae today cannot be produced for less than thousands of dollars per kilogram.

A critical component of the manuscript is the economics, it is in the title, yet the economics is lacking. The economics as presented are not complete. The work does not include the assumptions associated with the production costs. The economic work has minimal impact as it is done at small scale and with artificial lighting.

Further, microalgal systems will never be grown under a controlled light source. It will be cultivated outdoors where the light intensity cannot be controlled.

Reviewer #3 (Remarks to the Author):

The paper "Harnessing C/N balance to overcome the potential conflicts in economic microalgal production", by Han Sun et al, was revised according the suggestions.

I have some more comments:

The first section of the results ("carbon distribution...", lines 130-144) reports some experimental outcomes that are very well known in literature, but no comparison/discussion with available literature was done.

I still have some doubts about the problem of CO₂. As I understand, no external CO₂ was provided, and experiments were carried out in flasks with glucose addition. Without decreasing the impact of the results obtained, the discussion should clearly mention the implications of working under mixotrophic conditions. In particular, many comments about the photosynthetic efficiency, acclimation of pigments and Rubisco activity should be discussed considering the strong limitation by CO₂.

The absence of CO₂, and the presence of organic carbon, have also an effect on oxygen production/consumption, that of course may lead to inhibition if not properly controlled. Under high light, and in CO₂ limitation, photorespiration may also occur. These points should be at least discussed.

Reviewer #1 (Remarks to the Author):

Unless the authors include growth infrastructure and do a discounted cash flow rate of return analysis the economics in the paper are not of value. Statements like “Our findings provide a new orientation to maximize biomass productivity and value that makes microalgal production economically viable in various applications.” Are not accurate. It is a falsity as the economic viability cannot be assessed strictly by understanding the value of the biomass. Microalgae today cannot be produced for less than thousands of dollars per kilogram.

Response: Thank you for the comment. We have removed the economic analysis in the manuscript and supplementary material. Then the expression was revised in the manuscript. Please see the title, line 33-36 and 603 in red, and supplementary material.

A critical component of the manuscript is the economics, it is in the title, yet the economics is lacking. The economics as presented are not complete. The work does not include the assumptions associated with the production costs. The economic work has minimal impact as it is done at small scale and with artificial lighting.

Response: Thank you for the comment. The part of economics was removed in the manuscript and supplementary material for the accurate description, as our economic analysis considered the operational cost, not included the capital investment of the system integrated into a cash flow system. In addition, the economical in title was also

removed.

Further, microalgal systems will never be grown under a controlled light source. It will be cultivated outdoors where the light intensity cannot be controlled.

Response: Thank you for the comment. The sunlight outdoor is usually at a high intensity. Therefore, combined with feeding carbon source, the high light intensity could be used efficiently to increase the productivity of high-value products. We added the significance in manuscript. Please see line 424-427 in red.

Reviewer #3 (Remarks to the Author):

The paper “Harnessing C/N balance to overcome the potential conflicts in economic microalgal production”, by Han Sun et al, was revised according to the suggestions.

I have some more comments:

The first section of the results (“carbon distribution...”, lines 130-144) reports some experimental outcomes that are very well known in literature, but no comparison/discussion with available literature was done.

Response: Thank you for the comment. We added the comparison and discussion in this section. Please see line 135-137, 140-141, 155-157, 165-167 and 194-196 in red.

I still have some doubts about the problem of CO₂. As I understand, no external CO₂ was provided, and experiments were carried out in flasks with glucose addition.

Without decreasing the impact of the results obtained, the discussion should clearly

mention the implications of working under mixotrophic conditions. In particular, many comments about the photosynthetic efficiency, acclimation of pigments and Rubisco activity should be discussed considering the strong limitation by CO₂.

Response: Thank you for the comment. We added the implications of working under mixotrophic conditions and added the consideration of strong limitation by CO₂, when discussed the results. Please see line 132, 167-169, 178-179, 184, 197-200 and 417 in red.

The absence of CO₂, and the presence of organic carbon, have also an effect on oxygen production/consumption, that of course may lead to inhibition if not properly controlled. Under high light, and in CO₂ limitation, photorespiration may also occur. These points should be at least discussed.

Response: Thank you for the comment. We added the discussion. Please see line 169-171, 192-193, 197-200, 416-418 and 420-423 in red.